# Analytical Performance of the Commercial MucorGenius^®^ Assay as Compared to an In-House qPCR Assay to Detect Mucorales DNA in Serum Specimens

**DOI:** 10.3390/jof8080786

**Published:** 2022-07-27

**Authors:** Théo Ghelfenstein-Ferreira, Laura Verdurme, Alexandre Alanio

**Affiliations:** 1Laboratoire de Parasitologie-Mycologie, AP-HP, Groupe Hospitalier Saint-Louis-Lariboisière-Fernand-Widal, 75010 Paris, France; theo.ferreira@aphp.fr; 2Laboratoire Cerba, Saint-Ouen-l’Aumône, 95310 Paris, France; laura.verdurme@lab-cerba.com; 3Institut Pasteur, Université Paris Cité, Centre National de la Recherche Scientifique (CNRS), Molecular Mycology Unit, Unité Mixte de Recherche UMR2000, CEDEX 15, 75724 Paris, France

**Keywords:** mucormycosis, Mucorales, qPCR, molecular marker, serum, cell-free DNA

## Abstract

Standardized, reproducible and validated Mucorales quantitative PCR (qPCR) assays are needed in the context of routine testing in diagnostic labs. We, therefore, compared the commercial MucorGenius^®^ assay (PathoNostics, Maastricht) targeting five genera of Mucorales to our in-house qPCR targeting *Rhizomucor* spp., *Lichtheimia* spp. and *Mucor/Rhizopus* spp. To assess their analytical sensitivity, 25 frozen leftover serum specimens, which had already tested positive based on our in-house assay, were selected. These sera were from 15 patients with probable or proven mucormycosis. For analytical specificity, 0.5 pg from 15 purified fungal DNAs from nine different Mucorales genera were spiked into pooled qPCR-negative leftover serum specimens. All samples were tested in parallel with both assays and the quantitative cycles (Cq) were compared. A total of 13/25 (52%) serum samples were amplified by one of the two assays with only four of them detected with the MucorGenius^®^ assay. In spiked specimens, all targeted strains were successfully amplified by our in-house qPCR. The MucorGenius^®^ assay was not able to detect *Lichtheimia* *corymbifera* but successfully amplified all other species targeted by the kit and two additional non-targeted species (*Syncephalastrum* *monosporum* and *Saksenaea* *vasiformis*). The MucorGenius^®^ assay showed lower analytical sensitivity compared to our in-house assay. Indeed, the MucorGenius^®^ assay amplified more species, as expected, but showed a decreased detection of the frequent species *Lichtheimia* *corymbifera*.

## 1. Introduction

Mucormycosis is an invasive fungal infection caused by fungi of the order Mucorales [1]. In France, between 2012 and 2018, a total of 10,886 invasive fungal diseases were recorded from the French RESSIF Network. Among these invasive fungal diseases, 15% were invasive aspergillosis and 3% were mucormycosis (*n* = 314) [2]. Early initiation of adequate antifungal therapy within five days of diagnosis is associated with a reduction in mortality (83% versus 49% survival) [3]. Unfortunately, early diagnosis is hampered by the low sensitivity of routine diagnostic tools such as direct examination, histology and culture. Moreover, there are no commercially available antigenic or antibody assays for mucormycosis [4]. Indeed, the two routinely used ß-D-glucan or galactomannan antigen assays are not reliable for Mucorales infections. As far as we know, Mucorales do not expose gluconic cell wall sugars on the surface of their hyphae [5]. Therefore, microbiological diagnosis relies on microscopic examination and the culture of samples from the infection site [1]. However, quantitative PCR (qPCR) assays for detecting serum Mucorales DNA have shown very promising results for both assessing the diagnosis and anticipating the diagnosis compared to direct examination and culture [6,7]. The in-house assay reported is based on three different qPCR reactions which target the 18S rDNA gene of four genera, *Rhizomucor* spp., *Lichtheimia* spp. and *Mucor*/*Rhizopus* spp., including the most frequent species involved in human pathology in the United States and France [8,9]. When implemented as a screening tool in critically ill burn patients, this assay led to a 50% reduction in mortality when treatment was targeted by positive qPCR results [10]. Therefore, to allow easier access and implementation in non-expert centers, a standardized commercial qPCR assay is necessary and some are already available (MycoGENIE^®^ Aspergillus-Mucorales spp., Ademtech; MucorGenius^®^, PathoNostics; Fungiplex^®^ Mucorales RUO, Brucker), although their clinical performance needs to be evaluated and compared to in-house assays.

The MucorGenius^®^ assay has been evaluated by Mercier et al. with blood samples (whole blood, serum or plasma) from 16 patients with positive Mucorales culture. The sensitivity was 75% considering the positive culture as the reference with no comparison with another qPCR assay [11]. We therefore compared the MucorGenius^®^ assay with our qPCR assay for the detection of circulating Mucorales DNA in human serum samples. Additionally, we tested a DNA panel of different Mucorales species for a detailed analytical specificity.

## 2. Materials and Methods

### 2.1. Clinical Serum Samples

Twenty-five frozen positive serum samples were selected from 15 patients diagnosed with probable or proven mucormycosis according to the European Organization for Research and Treatment of Cancer and the Mycoses Study Group Education and Research Consortium (EORTC/MSGERC) criteria [12]. The main characteristics of the patients are described in Appendix A. These serum samples were leftover material stored at −80 °C. These samples had all been tested qPCR-positive using our routine diagnosis procedure including DNA extraction using the Qiasymphony DSP virus/Pathogen Mini kit (Qiagen GmBH, Hilden, Germany) and a Qiasymphony apparatus (Qiagen GmBH, Hilden, Germany), elution in 85 µL [13], and amplification and detection using primers and probes previously described [14]. Briefly, each qPCR amplification was performed in a final volume of 25 µL reaction mixture including LightCycler 480 Probes Master (2×) (Roche Diagnostics GmbH, Mannheim, Germany), 0.3 µM of each primer, 0.1 µM of the probe, and 9 µL of sample extract. The qPCR program consisted of 50 cycles at 95 °C for 15 s and at 60 °C for 60 s each and was performed in a LightCycler 480 Real-Time PCR System (Roche Diagnostics GmbH, Mannheim, Germany) and analyzed using LightCycler 480 Software version 1.5. Each run contained a positive control and a negative extraction control. This process has been referred to as the Qiasymphony procedure for clarity in the present study.

After thawing, the 25 clinical serum samples were not extracted using our in-house procedure but using the Janus Chemagic 360^®^ system (Perkin-Elmer, Zaventem, Belgium). Briefly, 1 mL of serum was automatically distributed in a PCR plate where the lysis buffer was added. Nucleic acids were then extracted based on magnetic particles binding after different steps of washing allowing the elimination of cellular debris. Finally, all nucleic acids were eluted in 100 µL final volume after the separation of the nucleic acid-magnetic particle complexes. The qPCR was performed as recommended by the manufacturer’s instructions (Perkin-Elmer) on DNAs extracted as described above. Briefly, MucorGenius^®^ PCR mix (10 µL), internal control (M13 bacteriophage, 10 µL) and Taq polymerase (1 µL) were added to 5 µL of each sample extract. The PCR reaction was performed on a CFX96 instrument (BioRad, Marne-la-Coquette, France). The primer and probe sequences were designed to target and amplify the 28S ribosomal RNA of different Mucorales species (*Rhizopus* spp., *Mucor* spp., *Rhizomucor* spp., *Lichtheimia* spp. and *Cunninghamella* spp.) but were kept confidential by the manufacturer. Negative results were considered valid if the internal control was positive with quantification cycle (Cq) values ranging from 30 to 36. In parallel, 9 µL of elution from each of the 25 samples was used to perform a new in-house qPCR as described above. This process includes freezing and thawing and has been referred to as the Janus Chemagic procedure for clarity in the present study.

Each qPCR amplification was performed in duplicate.

### 2.2. Specificity Study Using Spiked Serum Samples

Fifteen mold species (12 Mucorales and 3 Ascomycetes) were obtained from the French National Reference Center of Invasive Mycoses and Antifungals (Appendix A). DNA was extracted from a 5-day-old PDA culture at 30°C, using NucleoMag Plant (Macherey-Nagel, Düren, Germany) and the semi-automated KingFisher Flex Magnetic Particle Processor (Thermo Fisher Scientific, Vantaa, Finland). The DNA was adjusted at 5 pg/mL. Then, 1 mL of pooled in-house qPCR-negative fresh serum samples was spiked with 100 µL of purified fungal DNA of each fungal species. The 15 spiked serum samples’ DNA was extracted as above using the Janus Chemagic 360^®^ system. The qPCR amplification was performed using the MucorGenius^®^ assay and the in-house qPCR as described above for the clinical samples. The in-house qPCR amplification was performed in duplicate.

### 2.3. Statistical Analyses

Data were analyzed with Prism 8 software (GraphPad Software, La Jolla, CA, USA). Median and interquartile ranges (Q1–Q3) were provided for variables with a non-gaussian distribution. Cq values for qPCR-positive results and for in-house qPCR assay and MucorGenius^®^ assay were compared. In the case of duplicate amplification use, the lowest values were used for comparison. The qPCR results were compared using the unpaired Mann–Whitney test or Wilcoxon matched-pairs signed rank test, when necessary. In case of negative amplification by qPCR, a Cq value of 50 cycles was applied, representing the highest number of PCR cycles performed. Significance was defined by a *p*-value of 0.05 using a two-tailed test.

### 2.4. Ethical Statement

The present study is a non-interventional retrospective study with no additional sampling for the patients beyond that needed for standard diagnostics and no impact on management. According to the French Health Public Law (CSP Art L1121-1.1), such protocols do not require approval by an ethics committee and are exempt from the otherwise mandatory informed consent requirements.

## 3. Results

### 3.1. Comparison of qPCR Assays Based on qPCR Positive Serum Samples from Probable and Proven Mucormycosis Patients

The in-house qPCR was unable to amplify DNA from 13 out of the 25 previously qPCR-positive thawed serum samples using the Janus Chemagic procedure. For the 12 persistently qPCR-positive samples, an increase in Cq value (median of differences [IQR]: 3.5 (2.0–4.6)) was observed with the Janus Chemagic procedure versus the Qiasymphony procedure (median [IQR]: 38.5 (37.8–39.8) and 35.3 (33.8–36.84), respectively; *p* < 0.001) (Figure 1A). The initial Cq values of the serum samples which turned negative were significantly higher than the initial Cq values of the serum samples remaining positive with the in-house qPCR (38.4 (37.0–39.2) and 35.3 (33.8–36.8), respectively; *p* = 0.002).

The analytical performance was evaluated on 12 qPCR negative and 13 qPCR positive specimens. Three out of 13 serum samples were positive with both assays. Nine out of 13 (69%) were positive only with the in-house qPCR assay and four out of 13 (31%) specimens were amplified with the MucorGenius^®^ assay including one that was not amplified with the in-house assay (Figure 1B). The MucorGenius^®^ assay harbored an analytical sensitivity of 25% compared to our in-house assay. The 12 negative serum samples were negative with both assays. The median [IQR] Cq values for the in-house qPCR (*n* = 12) and MucorGenius^®^ assay (*n* = 4) were 38.5 (37.8–39.6) and 38.0 (36.7–40.5), respectively (*p* = 0.75).

### 3.2. Comparison of qPCR Assays Based on Spiked Serum

Using the in-house qPCR assay all the species expected (see Section 2) to be amplified were successfully amplified at a final DNA concentration of 0.5 pg/mL. Thus, a Cq ≤ 38.0 was obtained for *Lichtheimia corymbifera* (Cq = 38.0), *Mucor indicus* (Cq = 34.7), *Rhizomucor pusillus* (Cq = 36.7), *Rhizopus arrhizus* var. *arrhizus* (Cq = 35.1), *Rhizopus arrhizus* var. *delemar* (Cq = 36.9) and *Rhizopus microsporus* (Cq = 34.7). On the other hand, the species not included in the primer design were not amplified (*Syncephalastrum monosporum*, *Syncephalastrum racemosum*, *Actinomucor elegans*, *Apophysomyces elegans*, *Saksenaea vasiformis* and *Cunninghamella bertholletiae*).

Using the MucorGenius^®^ assay, seven spiked samples including *Mucor indicus* (Cq = 34.0), *Rhizomucor pusillus* (Cq = 43.1), *Rhizopus arrhizus* var. *arrhizus* (Cq = 36.6), *Rhizopus* arrhizus var. *delemar* (Cq = 42.1), *Rhizopus microsporus* (Cq = 33.4) and *Cunninghamella bertholletiae* (Cq = 42.9) were amplified. Although not included in the panel kit, *Saksenaea vasiformis* (Cq = 40.1) and *Syncephalastrum monosporum* (Cq = 33.3) were amplified (Appendix A). One species included in the panel kit was not detected (*Lichtheimia corymbifera*). *Actinomucor elegans*, *Apophysomyces elegans* and *Syncephalastrum racemosum* were not amplified as expected.

Regarding the species amplified with both assays (Figure 2), the medians [IQR] of positive Cq values for the in-house qPCR and MucorGenius^®^ assay were 35.1 (34.7–36.8) and 36.6 (33.7–42.6), respectively (*p* = 0.31). No amplification was observed with both assays using spiked serum samples containing 0.5 pg/mL of *Aspergillus fumigatus*, *Aspergillus flavus* and *Penicillium digitatum* DNA.

## 4. Discussion

This comparative study showed that the in-house qPCR was more sensitive than the MucorGenious^®^ commercial assay. We worked with qPCR-positive serum specimens stored for 1–2 years that were thawed for the purpose of this study and submitted to a different extraction procedure than our routine procedure. After the second extraction procedure, 12 were non-detectable with both assays, and only 25% (3/12) of positive in-house qPCR samples were amplified using the MucorGenius^®^ assay.

Previously, the MucorGenius^®^ assay showed a strong correlation with in-house qPCR in respiratory samples [15]. However, the use of respiratory samples is not applicable to serum samples. This is partly due to the high fungal load in respiratory samples. For example, the median [IQR] Cq values in the study were 30.4 (26.9–34.9) in respiratory samples [15]. In contrast, the median (IQR) Cq values were 38.0 (36.7–40.5) in serum in the present study, as previously reported, reflecting a lower fungal load [7,16]. Comparative studies should therefore be done at the limit of detection of the qPCR assays studied to be able to detect differences. Indeed, it is more relevant to use very low fungal loads in comparison studies as the goal of these assays is to detect as early as possible Mucorales DNA in at-risk patients for an early and optimal therapeutic management of the patient. However, a persistent limitation of comparative studies with a rare disease is the low number of clinical samples available after the procedure has used part of the sample for routine diagnosis. Thus, the present study was carried out on a small number of leftover samples. Unfortunately, the presently used extraction protocol seems less efficient than the previously used one and almost half of the previously positive samples turned negative with this new protocol and the Cq values of the in-house qPCR performed during the present study were higher than those obtained at the initial diagnosis. In detail, samples were eluted in different volumes using initial and subsequent procedures (85 µL vs. 100 µL, respectively). Therefore, nucleic acid concentration could be different and impact the final amplification result. In the absence of prospective real-time comparison, we cannot completely eliminate that the freezing–thawing cycle was not responsible for this difference. The comparison of the assays should not be limited to the comparison of the amplification step and the specific primers only, but also include specific important steps such as the type of specimen, the fraction of the specimen to be used and the extraction method [17,18].

The lack of sensitivity of the MucorGenius^®^ assay could be explained by the choice of the qPCR target (i.e., 28S) and primers. However, Rocchi, S et al. have already shown, by evaluating four different Mucorales PCR assays targeting 18S rRNA, 28S rRNA and ITS2, a good inter-laboratory agreement, despite the considerable diversity of the methods used (26 different combinations) [19]. Some differences were obviously observed. The target, although supposed to be present in the same number of repeats (18S and 28S are repeated in tandem), could influence the performances of the different qPCR assays.

The specificity of the MucorGenius^®^ assay was evaluated using a panel of spiked serum. The MucorGenius^®^ assay successfully amplified six of the seven expected species in the spiked serum samples. *Lichtheimia corymbifera* included in the panel kit was undetectable. Rocchi, S et al. have already shown the lack of sensitivity of this kit for *Lichtheimia corymbifera* in a previous study [19], where the authors performed interlaboratory evaluations of qPCR assays for Mucorales to assess reproducibility and performance [7,20,21]. A total of 24 spiked sera, six samples comprising three serial dilutions of two different Mucorales DNAs (*Rhizomucor pusillus* and *Lichtheimia corymbifera*) in four medical laboratories, were tested. Three samples (one *Lichtheimia corymbifera* and two *Rhizomucor pusillus*) were missed with the MucorGenius^®^ assay. For this interlaboratory study, Rocchi, S et al. used serum spiked with DNAs extracted from culture (the concentrations of these samples ranged from 10–2 to 10–1 pg/mL (data provided by Rocchi, S. et al.). Although the MucorGenius^®^ assay was not designed for the detection of *Saksenaea* spp., *Apophysomyces* spp., *Actinomucor* spp. or *Syncephalastrum* spp., one of the two *Syncephalastrum* species and *Saksenaea vasiformis* DNA were amplified by the commercial kit, suggesting that the specificity of the kit is wider than advocated by the manufacturer. The above species might not have been evaluated previously. These additional species detected by the MucorGenius^®^ assay could be considered as an expansion of the qPCR detection spectrum of this assay. In particular, it is of interest for *Cunninghamella* spp. which can account for up to 9% of Mucorales infections in transplant recipients in a specific report [8].

## 5. Conclusions

In conclusion, the MucorGenius^®^ assay showed lower sensitivity in serum for the species targeted by the in-house qPCR, and more specifically for *Lichtheimia corymbifera*. This lack of sensitivity can be explained by the fact that designing assays for an increased number of species can lead to a loss in primer specificity, as already shown for *Aspergillus* qPCR [22]. On the other hand, targeting a single species ensures an improved specificity but at the risk of missing the diagnosis for non-targeted species. This is probably what we observe here, even if the secrecy of the primers sequence of the commercial kits does not guarantee it. Nevertheless, the spectrum of targeted species in the MucorGenius^®^ assay is broader than previously stated. Thus, the use of the MucorGenius^®^ assay might be indicated for the detection of species not included in the in-house qPCR according to the epidemiology, and or some clinical situation with positive direct examination without positive culture and negative in-house qPCR results. Although difficult to implement because of the rarity of mucormycosis, prospective comparative studies are necessary to promote the use of commercial kits.

## Figures and Tables

**Figure 1 jof-08-00786-f001:**
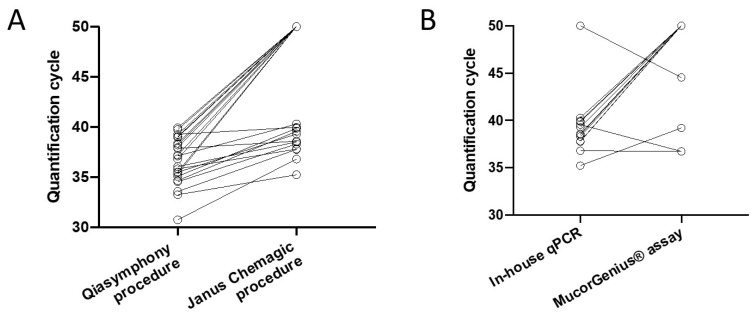
(**A**) Difference in quantification cycle (Cq) values between the initial result at diagnosis and the present study using a different DNA extraction protocol (Qiasymphony versus Janus Chemagic procedure) although using the same amplification protocol. Out of the 25 serum specimens, 13 specimens turned negative with the in-house qPCR and an increase in Cq value was observed with the Janus Chemagic procedure versus the Qiasymphony procedure (median [IQR]: 38.5 (37.8–39.8) and 35.3 (33.8–36.84), respectively; *p* < 0.001), Wilcoxon matched-pairs signed rank test). The initial Cq values of the serum samples that turned negative were significantly higher compared to the initial Cq values of the serum samples currently positive with the in-house qPCR (38.4 (37.0–39.2) and 35.3 (33.8–36.8), respectively; *p* = 0.002, unpaired Mann–Whitney test]. Negative results were attributed a Cq of 50. (**B**) Comparison of PCR assays using the Janus Chemagic procedure based on patient spec-imens. Thirteen serum specimens were amplified by at least one qPCR assay (in-house or Mucor-Genius^®^ qPCR). Negative results were attributed a Cq of 50.

**Figure 2 jof-08-00786-f002:**
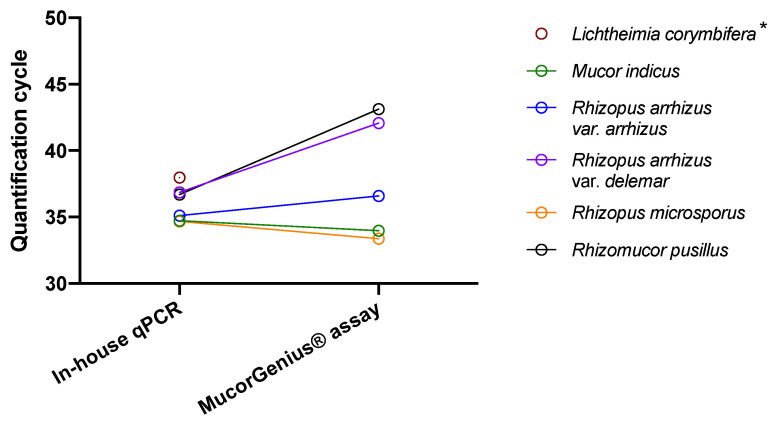
Comparison of qPCR assays based on spiked serum specimens detectable with both assays. The six sera spiked with DNA from fungi targeted by the two qPCR (*Lichtheimia corymbifera*, *Mucor indicus*, *Rhizopus arrhizus* var. *arrhizus*, *Rhizopus arrhizus* var. *delemar*, *Rhizopus microsporus* and *Rhizomucor pusillus*). * *Lichtheimia corymbifera* was not amplified with the MucorGenius^®^ assay.

## Data Availability

The data presented in this study are available in Appendix A (Appendix A).

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
