# Peer review of "Analytical Performance of the Commercial MucorGenius® Assay as Compared to an In-House qPCR Assay to Detect Mucorales DNA in Serum Specimens"

_jof, 2022, doi:10.3390/jof8080786_

Round 1
Reviewer 1 Report
In the present work, "Analytical performance of the commercial MucorGenius® assay as compared to an in-house qPCR assay to detect Mucorales DNA in serum specimens ", the authors are just comparing the clinical performance of one commercial kit available to diagnose mucormycosis, MucorGenius®, to their in-house qPCR assay.
The paper is well written, the statistical analyses nicely done. The introduction could provide a bit more information on a broader panorama, as in the recent years other methods, far easier and cheaper than qPCR have been published and found highly effective to diagnose mucormycosis infection, but no mention of that has been done. Indeed to find and provide an easy and effective method to early diagnose this illness it's auspicable, and such comparisons, when not available, provide interesting informations. I just consider that focusing only on one method of choice, qPCR based, without considering other options is a bit limiting and shorts-sighted. The same as keeping the focus on the 28S as target. Far more specific targets have been identified for Mcorales species and published as alternative and very valid options.
Minor points:
Line 68 : reported or described, perhaps one of the two is sufficient
Line 71: is 50 cycle a standard? It seems a lot for a qPCR
Author Response
Point 1: The introduction could provide a bit more information on a broader panorama, as in the recent years other methods, far easier and cheaper than qPCR have been published and found highly effective to diagnose mucormycosis infection, but no mention of that has been done. Indeed, to find and provide an easy and effective method to early diagnose this illness it's auspicable, and such comparisons, when not available, provide interesting informations. I just consider that focusing only on one method of choice, qPCR based, without considering other options is a bit limiting and shorts-sighted.
Reply: We thank reviewer 1 for this comment. Providing basic information on mucormycosis diagnosis, in addition to our sentence lines 35 – 37: “Moreover, there are no commercially available antigenic or antibody assays for mucormycosis [3]. Indeed, the two routinely used ß-D-glucan or galactomannan antigen assays are not reliable for Mucorales infections [4]”, is interesting for the reader.
As far as we know, qPCR is the best diagnostic tool for early and reliable diagnosis of mucormycosis. In serum samples, Millon et al. 2022 showed a sensitivity of >85% and earlier diagnosis up to 24 days before imaging (Millon et al. 2016). For example, Legrand et al., (2016) demonstrated reduced mortality in severely burned patients with invasive wound mycormycosis due to implementation of a systematic qPCR screening of plasma samples using the in-house qPCR, which led to earlier diagnosis and reduce mortality. The diagnosis of mucormycosis suffers from a lack of antigen detection, whether b-d-glucans or specific biomarkers, as well as a lack of antibody detection systems.
Action: we added the sentence lines 41 - 43 “Therefore, microbiological diagnosis relies on microscopic examination and the culture of samples from the infection site [1].”
Point 2: The same as keeping the focus on the 28S as target. Far more specific targets have been identified for Mucorales species and published as alternative and very valid options.
Reply: Our in-house assay targets 18S rRNA gene. Thank you for your comment, however the manufacturer keeps the primers secret which prevents us from discussing this aspect.
Action: We have added this paragraph regarding this point line 222 - 228: “The lack of sensitivity of the MucorGenius® assay could be explained by the choice of the qPCR target (i.e., 28S) and primers. However, Rocchi, S et al. have already shown, by evaluating 4 different Mucorales PCR assays targeting 18S rRNA, 28S rRNA and ITS2, a good inter-laboratory agreement despite the considerable diversity of the methods used (26 different combinations) [19]. Although some differences were obviously observed. The target, although supposed to be present in the same number of repeats (18S and 28S are repeated in tandem) could influence the performances of the different qPCR assays.”
Minor points:
Point 3: Line 68: reported or described, perhaps one of the two is sufficient
Action: Thank you, we modified the sentence for “using primers and probes previously described”. Line 72
Point 4: Line 71: is 50 cycle a standard? It seems a lot for a qPCR
Reply: Typically, we use from 45 to 50 cycles depending on the qPCR. Fifty cycles allow detection of very low fungal burden in cell-free DNA contained in serum samples. This has been published several times in Alanio et al. J Mol Diag, Frontiers 2017; Ghelfenstein-Ferreira et al. JCM 2020.
Reviewer 2 Report
The work of Ghelfenstein-Ferreira and cols. is an interesting study aimed to compare the commercial MucorGenius® assay targeting five genera of Mucorales with a proposed in-house qPCR targeting Rhizomucor spp., Lichtheimia spp. and Mucor/Rhizopus spp. In general, the study design, as well as the methods adopted were adequate and consistent with the planted objective. Moreover, the manuscript is well structured and delimitated. Despite its proper limitations, which are clearly mentioned in the manuscript, the work is informative.
Author Response
The work of Ghelfenstein-Ferreira and cols. is an interesting study aimed to compare the commercial MucorGenius® assay targeting five genera of Mucorales with a proposed in-house qPCR targeting Rhizomucor spp., Lichtheimia spp. and Mucor/Rhizopus spp. In general, the study design, as well as the methods adopted were adequate and consistent with the planted objective. Moreover, the manuscript is well structured and delimitated. Despite its proper limitations, which are clearly mentioned in the manuscript, the work is informative.
We gratefully thank Reviewer 2 for this comment and for taking the time to review our paper.
Reviewer 3 Report
Is Mucormycosis more abundant than aspergillosis? As I know galactomannan antigen assays show aspergillosis much more effectively.
1. They compared the commercial used MucorGenius assay and in-house qPCR assay to see which has higher analytical sensitivity.
2. I consider the topic is original and it is important to find more sensitive assay to detect Mucormycosis in serum.
3. It adds a new qPCR assay instead galactomannan antigen detection.
4. They do not to improve the methodology. They should compare the effectivity of galactomannan antigen detection in aspergillosis and mucormycosis.
5. The conclusions are consistent with the evidence and arguments.
6. The references are appropriate.
7. Figures help to interpret the results. These show the strength and weakness of the in-house assay.
Author Response
We thank reviewer for taking the time to review our paper.
Point 1: Is Mucormycosis more abundant than aspergillosis? As I know galactomannan antigen assays show aspergillosis much more effectively.
Action: We added the sentence to our introduction line 32 – 34: “In France, between 2012 and 2018, a total of 10,886 invasive fungal diseases were recorded from the French RESSIF Network. Among these invasive fungal diseases, 15% were invasive aspergillosis and 3% were mucormycosis (n=314) [2].”
Point 2: They do not to improve the methodology. They should compare the effectivity of galactomannan antigen detection in aspergillosis and mucormycosis.
Reply: Not all patients were screened for galactomannan antigen during hospitalization. Indeed as already well established (Garcia-Hermoso et al. 2015 Manual of Clinical Microbiology, 11th Edition, ch 121), galactomannan is not a useful biomarker for mucormycosis diagnosis, neither BDG. We do not use this biomarker as a routine diagnostic tool because galactomannan levels are not increased in mucormycosis, as Mucorales do not expose gluconic cell wall sugars on the surface of their hyphae (Pickering et al., 2005).
Action: We added the sentence line 40 – 41 to make this point clearer as “[…] Mucorales infections. As far as we know, Mucorales do not expose gluconic cell wall sugars on the surface of their hyphae [5].”